# A qualitative investigation of dental internships in Saudi Arabia: Exploring the experiences of dental interns

Mohammed Mahmoud Sarhan[1], Maram Ali M. Alwadi[2]*, Saleha Ali Alzahrani[3], Saad Mahrous Hasubah[4], Reem Hussain Alhammad[4], Ali Muhammed Alhussain[4], Leen Ahmed Qarras[4], Shadan Hani Sharbib[4]

1 Department of Preventive Dental Sciences, College of Dentistry, Taibah University, Medina, Saudi Arabia, 2 Department of Dental Health, College of Applied Medical Sciences, King Saud University, Riyadh, Saudi Arabia, 3 Pediatric Dentistry, Dental University Hospital, University Staff Clinics, King Saud University Medical City, Riyadh, Saudi Arabia, 4 College of Dentistry and Hospital, Taibah University, Medina, Saudi Arabia

* Malwadi@ksu.edu.sa

## Abstract

### Background

For dental graduates, internships are a vital transitional phase that gives them the invaluable opportunity to close the gap between their academic studies and the reality of professional dentistry. Research on dental internships remains limited and most of the existing studies focus on the clinical aspects of dental internships with little attention given to dental interns' experiences overall. This study aims to bridge this gap in the literature by gaining an in-depth understanding of Saudi dental graduates' range of experiences as dental interns.

### Methods

To achieve the research objective, this study adopted a qualitative approach. Using purposive and snowball sampling, the study recruited 23 dental interns from Riyadh Province, Saudi Arabia, who had completed at least nine months of their internship. Data was gathered across three months via diaries and virtual semi-structured interviews based on participants' preferences. The data was then analysed thematically using an inductive analytical strategy.

### Results

The data analysis revealed three major themes and four sub-themes regarding the experiences of dental interns. The core three themes were "activities", "autonomy" and "transitioning to a balanced life" whereby interns have the time and freedom to explore their interests, rekindle their social lives and focus on self-care, resulting in a better work-life balance.

### Conclusion

This study suggests that dental interns will benefit from the retention and strengthening of key internship activities such as research, community work and clinical rotations.

**Data Availability Statement:** All relevant data are within the manuscript.

**Funding:** This research project was supported by a grant from the "Research Center of the Female Scientific and Medical Colleges", Deanship of Scientific Research, King Saud University.

**Competing interests:** The authors have declared that no competing interests exist.

Additionally, the experiences of dental interns can be improved by encouraging interns to progress in their clinical training with a high level of autonomy. Also, due to the limited research in this area, further studies are required to improve our understanding of the lived experiences of dental interns and dental internships in general.

## Background

The focus of dental education is to provide learners with the knowledge and skills essential for effective oral healthcare, while also encouraging lifelong professional growth [1] to enable new dental practitioners to face the challenges of real-world dental practice [2]. Consequently, dental education programmes worldwide require dental graduates to complete an internship of a specific duration, during which they are exposed to clinical practice [3].

Internationally, a range of terms are used to describe dental internships [4–6]. For instance, in the UK, they were initially referred to as "vocational training for general dental practice" and are now called "foundational training" [4]. In South Africa, these internships are known as "compulsory community service" [5], while in Saudi Arabia they are simply "dental internships" [6].

Dental internships have several aims and objectives. Specifically, they aim to enhance graduates' general dentistry knowledge and clinical skills under supervision [6–8] and foster greater responsibility among interns regarding patient care and clinical competency [9]. They promote various abilities, including encouraging interns to participate in different activities such as research and community works, which improve related skills [10], motivate interns to pursue postgraduate study [11], support interns' personal growth and networking [12] and enhance interns' competitiveness for academic positions [13].

Several studies have examined the challenges interns face during their internships and the factors that influence their success [2, 14]. These challenges are the consequence of moving from the role of student to the role of dental practitioner [14]. The most common difficulties that interns face during this transitional stage are increased patient loads and limited time in which to deal with them, and greater responsibility for patient treatments and care [2, 14]. Interns also struggle with adapting to a new work environment and forming professional relationships with colleagues [2].

The success of interns as they transition from student to professional is influenced by factors such as the availability of supervisors who understand how to enhance interns' educational experiences [14]. Research in medicine has found that the availability of an effective supervisor and the engagement of trainees is vital to strengthening trainees' practice-based education and enabling them to tackle new responsibilities [15]. Therefore, clinical supervisors must exhibit professionalism, possess key theoretical and clinical knowledge and provide constructive feedback to enrich interns' learning experiences [14, 16]. Moreover, organisations can improve the clinical learning environment by updating facilities and providing additional learning materials, including chairs and space to study [17]. Notably, interns' interpersonal skills, personal characteristics and attitudes also significantly impact their ability to overcome internship challenges [18, 19].

Some studies examine the experiences of dental interns but only in terms of their clinical experiences [3, 14, 20, 21]. Ramalingam et al. [3], for example, explored dental interns' opinions regarding their experiences with dental implants in patients who are medically compromised. The authors found that 47% of the interns surveyed had some experience with dental

implants while 42% had direct experience of implanting them in medically-compromised patients. Dental interns' experiences after they completed their accident and emergency rotations were investigated by Shenoy et al. [20], who found that, despite being exposed to a range of cases and growing in confidence, the interns remained concerned about conducting emergency procedures. Further research [14] has been done to examine dental interns' perceptions of supervision and the learning environment. The study on this topic reported that both female and male interns were extremely satisfied with the training they received as interns.

On the other hand, few studies [2, 11] have examined dental interns' experiences beyond their clinical training, exploring their personal and social experiences as interns. Across these studies, only a few are qualitative [2]. Therefore, to bridge this gap in the literature, this study aims to qualitatively explore and gain an in-depth understanding of Saudi dental interns' range of experiences during their internships to identify potential areas for improvement and measures to enhance their overall experience.

## Research methodology

### Philosophical perspective

To describe our philosophical perspective clearly, we have followed the key elements outlined by Killam[22] in research philosophy. From an ontological perspective, we adopt a "relativist" position, which argues that there is no one reality but many realities depending on the meanings given to them. As a result, reality only exists with these associated meanings. Thus, this paper adopts the perspective that reality varies based on how an individual views it and the meanings they attach to it. Consequently, reality changes and adapts in different settings and contexts and in response to people's different experiences. Additionally, our ontological perspective forms the basis of our epistemological or knowledge position, specifically, what we consider to be knowledge and how we gain it. Given our belief that reality varies based on the setting and people's experiences, we have decided not to attempt to step back from the research and objectively measure the reality we are studying but instead engage with the study and the study participants to try to capture the meanings attached by the participants to truth so that we can understand the reality we are investigating. Our qualitative research methodology aligns with these ontological and knowledge positions as it is designed to gain a detailed comprehension of dental interns' real experiences by capturing the contexts, circumstances and meanings connected to dental internships by the participants.

### Research design

A qualitative research design was applied to complete the data collection and analysis in this research. This study is part of a larger research project designed to develop a thorough knowledge base on dental interns' experiences of their internships and the day-to-day reality of these internships.

### Research context

This study was conducted in Riyadh Province, Saudi Arabia, with research subjects recruited among dental interns who had earned bachelor's degrees in dentistry from two universities in Saudi Arabia.

### Sampling strategy and participant recruitment

Two sampling techniques, purposive and snowball sampling, were applied in this study. First, purposive sampling was used where dental interns a minimum of 9 months into their

12-month internship were asked to participate in the study. Then, during data collection, snowball sampling was used to recruit additional participants. Specifically, the initial participants (recruited through purposive sampling) were asked to suggest others who may be willing to participate in the research. These potential participants were then approached by the researchers at their workplace who invited them to join the study. The study was explained to all participants and their contact details were recorded so that interview times could be arranged. The participants were sent topic guides before the interviews to allow them to reflect on the issues to be addressed and take notes to remind them of the points they wanted to discuss in the interviews. Following Ritchie and Spencer [23], saturation point was reached after 23 participants were interviewed when the researchers determined that no new data or themes were emerging from the interviews. Thus, no further participants were required and the sample size of 23 participants was considered sufficient to produce a comprehensive understanding of dental interns' experiences.

## Research subjects/participants

A total of 23 dental interns, 8 men and 15 women, were recruited to participate in this research. All of the participants had begun their dentistry studies in 2016 and completed six years of dental college before they started their internships. Moreover, the participants had all completed a minimum of 9 months of their 12-month dental internships. The interns were in their mid-20s with 20 interns aged 24 and 3 aged 25. During the data collection process, 9 research subjects were interviewed and 14 submitted diaries.

## Data collection methods

Semi-structured interviews and diaries were used to collect the data needed in this study. The method used depended on the participants' preferences and which method they believed would allow them to provide more complete data. All of the participants who chose to complete the interviews asked to complete them online rather than face-to-face. The interviews were primarily conducted in Arabic but the interviewees had the option to use English if they wished. The interviews were 30 to 45 minutes in length. The participants who chose to submit diaries all asked to receive the prompts for their diary entries in English and responded in English. The data collection process took place across the three months from March 5 to June 7, 2023.

## Data collection instruments

Either Zoom or Microsoft Teams was used to conduct the interviews. WhatsApp was used to send the diary prompts to the research subjects who chose to submit a diary as it was accessible and easy to update. Microsoft Word was used to compose the diary prompts before they were sent to participants via WhatsApp. The researchers leveraged their academic experience to design a topic guide for the interviews that would facilitate the free discussion of the interviewees' experiences and maximize the value of the data obtained through the interviews. The guide included three open-ended questions that referenced the educational, clinical, and social aspects of the internships and allowed the interviewees to discuss anything they felt was pertinent to the objectives of the research. The three open-ended questions were:

1. What are the educational, social, and clinical aspects of your internship?

2. What examples do you have of the practical differences between the educational, social, and clinical aspects of your internship and those of your undergraduate studies?

3. What educational, social, and/or clinical practices have you encountered during your internship that you did not encounter during your undergraduate studies?

When necessary during the interviews, the interviewer asked the interviewee to provide further detail about and expand upon their answers. The researchers used the responses from the first few interviews to refine the interview topic guide and introduce additional questions. All interviews were recorded on an audio recording device.

## Data processing and analysis

The researchers transcribed the recordings of the interviews and then translated them into English. As the diaries were in English, they arrived ready for analysis. Data management and security protocols, including the prevention of unauthorized access to the transcripts and anonymization of the transcripts, were used to protect the research subjects' confidentiality.

Before the data was analyzed the researchers agreed to employ thematic analysis following the steps suggested by Braun and Clarke [24–26]. Specifically, the researchers would familiarize themselves with the data before establishing an initial set of codes, identifying themes, reviewing these themes, naming the final themes identified, and composing the research report.

Thus, as a starting point, the researchers read the interviews, beginning with the first interview, and noted key data as they became increasingly familiar with the data collected. The researchers then established an initial set of codes by identifying the patterns that were emerging from the data and naming these patterns. The codes were refined as they were progressively grouped into themes. As additional data was analyzed, the themes identified were reviewed and adapted as appropriate. The researchers gave each theme a brief, descriptive title that illustrated the elements covered by that theme. Finally, the themes were used to compose the research report in which each theme was comprehensively analyzed and the findings of the research were examined.

Braun and Clarke [24] recommend that researchers who conduct thematic analysis can better clarify their direction in analysis by determining, for example, what is defined as a theme and whether an inductive or theoretical approach is used in thematic analysis or realist or constructionist analysis is most appropriate. It is also necessary for the researchers to determine whether they are identifying semantic or latent themes.

The themes identified in this study are seen as reflective of the meaning of the data gathered, as per Braun and Clarke [24] Nevertheless, the frequency of a theme's appearance in the data is not necessarily understood to reflect the importance of that theme, in line with Braun and Clarke [24] and Nowell et al. [27]. Inductive thematic analysis was applied in this research whereby the themes were taken from the data collected and were not prepared based on a literature review or review of existing theory. This form of analysis operates in conjunction with the realist analytical approach as the researchers view the data collected as reflective of the research subjects' experiences and not of any social or cultural pressures. Moreover, the researchers determined that the themes of this research were semantic as the analysis concerned the data's direct meanings. The social and cultural factors that may have affected the participants' responses and any other elements outside of direct meaning were not examined in this study.

## Verification of the trustworthiness and credibility of the data analysis

Triangulation, member-checking, and an audit trail were employed to verify the data analysis's trustworthiness and credibility. Triangulation was achieved through the use of interviews and

diaries as sources of data while member-checking ensured that the interpretations of the researchers and the findings of the research were both accurate and complete. Furthermore, an audit trail was maintained to establish a record of the data collection and analysis processes and the researchers' decision-making.

## Reflexivity

We believe that the researchers share two traits that have the potential to impact the study. To begin, the researchers have experience in dentistry and, as part of their studies, some of them completed a year-long dental internship. The researchers' personal experience with these internships gives them a deeper comprehension of the different elements of these internships. Additionally, the researchers have comprehensively researched this area and have thus gained numerous insights into dental interns' varied experiences.

Due to these two factors, a degree of subjectivity may have been introduced into the study and the researchers' interpretations of the collected data could have been affected by their own internship experiences. As an example, the researchers may have unconsciously focused on and highlighted particular themes that resonated with them and paid less attention to other themes. Moreover, the participants themselves could have influenced the researchers' data interpretations. The participants will have different experiences as interns and these subjective experiences could result in them unconsciously concentrating on particular subjects or questions more than others.

## Ethical considerations

Throughout this study, the research subjects' privacy and rights were thoroughly considered. Ethical approval for the research was granted by the Institutional Review Board of Taibah University (TUCDREC/270223). The confidentiality and anonymity of the participants were protected through the use of pseudo-anonymization procedures throughout the research. Specifically, the quotes used in the results section were attributed to unique codes such as P1, P2, P3 and so on. These codes were used in place of the participants' names to protect their anonymity and confidentiality. Before agreeing to participate in this study, the dental interns targeted were sent an information sheet on the study and had the opportunity to ask questions over the following week. The objectives of the research were explained again during the interviews and the interviewees were able to ask questions if they wished. The research subjects signed a consent form indicating their written consent to participate in the study after the research had been explained to them. This manuscript applies O'Brien's [28] guidance on how to present qualitative research as applicable.

## Research results

The analysis of the data obtained from the semi-structured interviews and diaries revealed several key themes and sub-themes. Of these, the most important were identified as "activities," "autonomy," and "transitioning to a balanced life."

## Activities

The key theme of "activities" emerged from the data collected from the semi-structured interviews and diaries. This theme covers various aspects of the dental interns' experiences during their internships and contains the sub-themes of "clinical rotations," "research," "community work," and "workshops and conferences."

## Clinical rotations

The sub-theme of "clinical rotations" investigates the experiences of dental interns in their clinical rotations. It covers the dynamic nature of dental internships as reflected in the interns' involvement in practical training across different dental centers and their immersion in a vibrant and changing learning environment. Moreover, this sub-theme highlights the significant impact that clinical rotations have not only on the professional development of interns but also on their social and personal growth. Clinical rotations allow interns to acquire and then refine important clinical oral healthcare skills and engage in a process of self-discovery through which they gain insight into their weaknesses and strengths. Thus, through clinical rotations, interns develop their clinical expertise, expand their knowledge, and develop their social and soft skills. One participant explained the importance of clinical rotations to their professional, personal, and social development in the following terms:

> "My internship was divided into three parts and every part was in a clinical center. Each center offered its own enjoyable experiences. I developed my time management skills and social skills especially at one of those centers as it was a completely new environment for me. I established several relationships with colleagues and doctors there. It was mandatory to practice at the first clinical center where you work in the dental emergency department. This was a new experience for me. I learned from the supervisors there, they have different specialties, and they helped me to master new clinical techniques I had not learned before. There was a strong spirit of collaboration." (P4)

From this participant, we learn that dental interns participate in training programs across a range of healthcare centers and gain a variety of benefits from these experiences. In particular, this participant highlights the improvement in their time management skills as part of their personal development. Moreover, in terms of their social development, this participant points out that they built meaningful relationships during their rotations. Finally, clinical rotations significantly enhanced this participant's clinical abilities and helped them to gain new abilities and learn new oral healthcare techniques. A further participant discussed their experiences of clinical rotations, stating:

> "Internships allow us to practice our clinical skills at different clinical centers. This means that we are in contact with and deal with a lot of people. Through your time at the centers, you can develop yourselves, find your limitations, and find your strengths. One of the centers I went to specializes in treating medically compromised patients. This was a great chance to be exposed to those kinds of patients and one I may not have again, that's why I chose this center as it has helped me to deal with medically compromised patients." (P3)

Again, this participant felt that their clinical rotations significantly improved their clinical abilities, particularly concerning the treatment of medically compromised patients. This participant emphasizes that they were exposed to valuable new oral health information about this group of patients. Socially, clinical rotations gave this participant the chance to interact with a diverse range of people and allowed them to discover more about themselves, in particular, their weaknesses and strengths.

## Research

The qualitative analysis of the theme of "activities" also identified "research" as an important sub-theme that addresses dental interns' involvement in research activities. Under this sub-

theme, participants discussed their engagement in research during their internships and elucidated the significant impact that their research work had on their professional and personal growth. Through research, interns embarked on a process of self-discovery, learning, and multifaceted development. They were able to explore new areas of learning, develop their research abilities, and improve their understanding of their research capabilities. Thus, the testimonies presented in this section showcase the participants' reflections on the ways in which research broadened their academic horizons and their self-awareness. To illustrate, one participant discussed the transformative impact of the research work they undertook during their dental internship:

> "For the research element, we decided to tackle something different during the internship [compared with my undergraduate degree]. I had never conducted a randomized clinical trial before, but we started one during the internship. We are now at the sampling stage. It's a very difficult step. This research is very new to us but because we have free time, we have enough time to recruit participants. Honestly, I have been learning from this experience; I have learned a lot; I have discovered myself. It's very nice to discover yourself and know what you can do in the limited timeframe of an internship. Although we have not finished this research yet, I am completely satisfied with what we have achieved." (P11)

This participant gained a great deal from their research experiences as an intern. These benefits included gaining new knowledge of both oral healthcare and their character and capacity. The value of engaging in research activities during a dental internship was also discussed by Participant 19, who stated:

> "In the internship, [research] is a requirement. We started our research with a very good doctor [as our supervisor]. I chose him. We are still working on this project. It's been a nice experience. We entered the lab, we did the experiments by ourselves, and the doctor was only there if we needed him. I enjoyed the experience. The group is very nice and the doctor is as well. He gives us guidance. He knows our level and gives us guidance accordingly. We have enjoyed the experience very much. I started another research project with the same group and we will publish the results of this second research project right before the start of postgraduate applications. The first research project was taking too much time so we decided to make it simply a learning experience to satisfy the requirements of the internship." (P19)

The guidance that this participant received from their supervisor was a pivotal learning experience for them. It enabled them to increase their knowledge base and develop their writing skills and experience with the publication of academic work, attributes that will help advance their career.

## Community work

The data analysis also identified the sub-theme of "community work" as central to the experience of dental interns. According to the data, dental interns are active participants in community activities such as educating school students about oral health through school visits. Moreover, the research participants explained that they even brought members of the community to the university hospital so they could access the dental services they needed. The data reveals that dental interns help to promote oral health awareness through their work with the community and deliver vital oral healthcare to community members. One participant

described the positive impact of the community work they undertook as a dental intern, offering a valuable perspective on this sub-theme:

> "We have to complete a community project as a requirement. You have to visit schools, deliver lectures, and provide information about oral health and such. I did one and visited a school. It was a great opportunity and a fun experience." (P12)

Alongside the insights offered by this participant into the sorts of work dental interns do in the community to educate the public about oral health, a further participant stated:

> "It is a requirement to conduct community work. We went to a school, you know, we gave a presentation. Although the requirement is to do one community activity, some of us did several. We gave a presentation [about oral health] and we showed them how to brush their teeth and things like that. One of the community service activities that I am doing right now is with a charitable organization. Through them, I have been able to reach some patients [who are in need of oral treatment] and bring them to the university hospital for [free] treatment." (P7)

This participant again discusses the work they do to educate the public about oral health and also mentions the support they offer to members of the community in need of dental services, helping them to access vital dental healthcare.

### Workshops and conferences

The identification of this sub-theme stresses the important role that workshops and conferences play in the experiences of dental interns. Specifically, workshops and conferences provide interns with vital learning opportunities and enhance the overall impact of their training. The data reveals that dental interns attend workshops and conferences to expand their dentistry and oral healthcare knowledge. The events they attend are on a broad range of topics and they benefit from hearing from accomplished professionals and experts. The testimonies below typify the kinds of experiences the research subjects had during their internships and provide insights into their opinions and experiences with workshops and conferences as interns in terms of their professional growth:

> "For the workshops, all the workshops that I have attended during my internship have been on themes that relate to my future specialty, periodontics, or are about diagnosis or some about prosthodontics. In general, the workshops helped me to improve the dental care I provide overall and cover all sorts of things. They were good. The last conferences I have attended have all featured international speakers. I have enjoyed benefiting from their insights and that's why I attended." (P5)

> "Throughout my internship, I was able to attend several workshops and conferences. I was not able to do that during my undergraduate studies." (P6)

### Autonomy

The research subjects expressed the newfound feeling of autonomy they experienced as interns in their diaries and interviews, leading to the identification of the theme of "autonomy" from the data. From the data, we learn that the activities that dental interns engage in, especially research and clinical practice, increase their self-confidence and self-reliance and result in

behaviors that show the level of autonomy dental interns have as professional dental practitioners. The qualitative analysis shows that, over time, the participants relied less and less on their supervisors when practicing in their clinics. The two following testimonies help to illustrate the perspectives of dental interns on the autonomy that they enjoyed during their internships regarding their clinical practice:

> "There is always supervision just in case we need it, but [the supervisors] let you work. They tell you to "go ahead and work and don't call me." As an undergraduate, if I were doing a restoration, for example, after each step I had to call the doctor; after the caries removal, after the restoration placement. Now I call her only when I have a problem." (P5)

> "I've started to feel more independent when treating my patients and have a greater sense of self-confidence and responsibility regardless of the presence of a supervisor." (P16)

The research completed by dental interns also gives rise to the experience of autonomy. The participants explained that their reliance on research supervisors decreased significantly during their internship, revealing a heightened level of autonomy and self-reliance when undertaking research. One participant discussed the independence they were afforded when they engaged in research as a dental intern:

> "We entered the lab, we did the experiments by ourselves, and the doctor was only there if we needed him. I enjoyed the experience. The group is very nice and the doctor is as well." (P19)

### Transition to a balanced life

A key result of the qualitative analysis of the data was the identification of the theme "transition to a balanced life," which reflects the importance of the achievement of a balanced lifestyle for dental interns. Dental interns experience a degree of freedom during their internships that they were not given as undergraduate students. This freedom allows them to explore their interests, engage in self-care, and spend time nurturing their relationships. According to the participants, during their internships, they were able to change their day-to-day routines and devote more time to their personal lives, improving their work-life balance. In response to this newfound freedom, some participants went on journeys of self-discovery and others rekindled relationships that had suffered because of the demands of their academic studies. The "transition to a balanced life" theme underlines the transformational aspect of dental internships on interns' lives. In sum, dental interns are free to pursue a more satisfying and harmonious lifestyle that is not dominated solely by professional responsibilities and considerations. The following testimonies clarify how the participants viewed and navigated the transition from being an overworked undergraduate to a dental intern:

> "I underwent surgery during my internship. I had the time. Before, as an undergraduate, I had no time to recover [from surgery] but it was the first thing I did in my internship. My friend also had surgery during her internship. We have put off doing a bunch of things until our internships. During my internship, I have also returned to seeing my school friends and we have created a new routine: we meet every other Saturday. It's official. During my undergraduate studies, I could not do that. . .as an intern, I have free time and no exams. When I am back from work I have nothing to do, so I take care of myself, I have a social life." (P1)

"Now, as we are approaching the end of the internship and we have finished everything, we have free time. We just go to the gym, travel, socialize, we have plenty of time. Actually, we treated ourselves this year." (P4)

## Discussion

The present work is the first qualitative study to explore the experiences of Saudi Arabian dental interns. The three main themes that emerged from the data analysis were "activities," "autonomy," and "transition to a balanced life." Under the theme of "activities," it was found that dental interns are involved in a variety of activities reflected in the sub-themes of "clinical rotations," "research," "community work," and "workshops and conferences." Specifically, clinical rotations offer interns practical training opportunities at various healthcare centers and, in this way, support interns' professional, social, and personal development. Similarly, the research that the dental interns participated in during their internships supported their professional and personal growth as did their involvement in community work, which included educating school students about oral healthcare and helping community members to access vital dental services. Finally, workshops and conferences were found to be a vital part of the participants' dental internships and instrumental to their professional development. The data also revealed the central theme of "autonomy", which reflects the independence and greater self-reliance in decision-making that the participants experienced during their internships. The participants manifested this autonomy through their research activities and clinical work. A third core theme, "transition to a balanced life," also emerged under which the participants expressed the feeling of freedom that they experienced during their internships. As interns, they were able to participate in a range of activities that they did not have time for as undergraduates. This change in their daily lives allowed them to pursue self-care and nurture their interests and friendships.

Currently, there is limited research available on dental internships [2, 3, 11, 14, 20, 21]. The lack of pertinent research in this area is striking given the expansive and dynamic nature of internships and their many dimensions covering the clinical, personal, and social, as well as the domains of challenges, advantages, perceptions, experiences, concerns, and so on. Moreover, the literature in this area tends to adopt distinct research directions and viewpoints. To illustrate, Alenezi [14] employed a questionnaire to investigate dental interns' perceptions of their learning environment and the supervision they were receiving, while Ramalingam et al. [3] and Shenoy et al. [20] examined dental interns' clinical experiences. Specifically, Ramalingam et al. [3] explored interns' perceptions of implant placement in medically compromised patients, and Shenoy et al. [20] investigated interns' perceptions, concerns, and experiences following accident and emergency rotations. Research by Ralph et al. [11] and Ali et al. [2] was broader in scope. Ralph et al. [11] compared the experiences of dentists who began their dentistry careers as vocational dental practitioners, that is, those who had completed an internship before embarking on their professional life, with those who started as associates, that is, those who had not completed an internship before entering professional dentistry. Additionally, Ali et al. [2] investigated the transition of new dental graduates to professional life to comprehensively elucidate the difficulties and advantages of completing a mentored year in a general dental practice setting.

This research explores dental interns' general experiences during their internships. Given the differences between the direction, methods, and findings of this research and those of the studies mentioned above [2, 3, 11, 14, 20, 21], it is difficult to draw direct comparisons between findings. Nonetheless, it is important to note that there are similarities between this research

and that undertaken by Ralph et al. [11] and Ali et al. [2] in that all three studies explore the general experiences of dental interns and not specific elements of internships, for example, clinical practice. Consequently, the three studies may share key findings that are worth examining more closely.

The findings of this work and that of Ralph et al. [11] and Ali et al. [2] reflect the broad scope of these studies and the assumption that dental internships involve a range of dimensions beyond the clinical, such as the personal and social. Nonetheless, the specifics of these personal, social, and clinical dimensions differ across the studies. For example, Ralph et al. [11] reported that many of their study participants considered their clinical experiences to be the most valuable component of their internships. Through these clinical experiences, the interns noted that they could improve their clinical decision-making, use new techniques at their leisure, build practical treatment interventions, and participate in combined complex treatment sessions with educators. The findings of the present work similarly reveal the important role played by clinical rotations in improving dental interns' clinical proficiency and expanding their knowledge but also find that clinic work nurtures interns' personal and social skills and encourages them to engage in a process of self-discovery that can give them greater insight into their weaknesses and strengths.

In terms of the social dimension of dental internships, a substantial proportion of the participants in the Ralph et al. [11] study considered a key advantage of internships to be the chance they afforded interns to interact with and exchange experiences with their peers, which provided the interns with valuable support. Moreover, a significant number of participants highlighted the value of receiving assistance from dental colleagues and mentors. The present research, however, identified a significant social element to the interns' experiences that revealed that dental interns build relationships with colleagues through clinical rotations and dental internships give interns the time to rekindle old friendships.

The investigation of the personal dimension of dental internships in the existing literature [11, 12] has revealed that dental internships can be negative experiences for interns. To illustrate, research [11] has reported that a significant number of interns who completed their internships experienced personal issues such as financial pressures and isolation. Further research [2] has reported that the transitional stage of an internship is considered by dental graduates to be both a stressful and challenging phase in their education. In contrast to these findings, the research subjects in the present work reported that their internships were a positive experience. The findings of this study highlight the importance of engaging in a range of activities, which support interns' professional and personal development and ready them for the rigors of a career in dentistry. Additionally, this study finds that internships provoke a feeling of freedom and autonomy in dental interns and introduce them to a new lifestyle that gives them the time to tend to their well-being and engage in activities that they neglected as undergraduates due to the demands of their studies. Specifically, dental internships allow interns the freedom to devote time to their interests, old friends, and self-care. The positive experience of the interns surveyed in this research may be due to the difference in the structure of dental internships in Saudi Arabia to those investigated in other contexts. However, the focus of this research, its methodology, and the interview prompts did not concentrate on the difficulties and challenges that the research subjects faced during their internships. Consequently, the negative experiences of the participants may not have been highlighted by the data collection process. The differences in interview prompts, methodologies, and so on between the studies can also explain other variations in the results concerning the clinical and social dimensions of dental internships.

Given that variations in methodology used played a significant role in producing different outcomes in dental internship literature, we deemed it essential to closely examine this aspect

in order to obtain valuable insights that contribute to this field. This study thus includes a detailed examination of some aspects of the dental internship literature that covers both qualitative and quantitative research [2, 3, 11, 14, 20, 21]. For example, Ramalingam et al. [3] and Ralph et al. [11] employ quantitative research methodologies in contrast to Ali et al. [2], Shenoy et al. [20], and Sebastian et al. [21] who all adopt qualitative methods. The different methodologies used identify a varied array of findings on diverse elements, significantly enhancing the research in this area and helping to build a thorough body of knowledge on dental internships.

Ramalingam et al. [3] and Shenoy et al. [20] are good examples of the impact of different methodological approaches on a study's findings. Both these studies explored dental interns' experiences following the completion of "clinical training," specifically, following the completion of a rotation on implant placement in medically compromised patients in Ramalingam et al. [3] and following the completion of an accident and emergency rotation in Shenoy et al. [20].

To illustrate, Ramalingam et al. [3] reported that 47% of the participants had worked with dental implants and 42% had placed implants in patients who were medically compromised. In contrast, Shenoy et al. [20] who collected qualitative data from interviews, reported that, following their rotations, the interns were still concerned about being unfamiliar with emergency procedures; the rotation had improved the interns' confidence and they had learned from the different medical cases they encountered; the interns remained worried about not having sufficient knowledge; the interns felt that medical rather than dental interns were the focus of accident and emergency rotations and this element of their training was not sufficiently long. As a result, the participants of the Shenoy et al. [20] study suggested that dental teachers should be present during accident and emergency rotations and they needed more specialized training and a better introduction to emergency dental practice. Moreover, the interns expressed their belief that hands-on practical experience was more valuable than earning an emergency dental care certificate.

Both approaches to research, the numerical approach (quantitative research) and the study of the nature of phenomena (qualitative research), are extremely valuable in the study of dental interns' experiences. The quantitative data reported in research such as that done by Ramalingam et al. [3] provide figures that show the concrete practice of dental interns. In contrast, the qualitative data reported by Shenoy et al. [20] allows us to better understand dental interns' perceptions, emotions, and attitudes. Together, these two sets of data provide a comprehensive overview of dental internships. Put differently, through quantitative research, we understand the "what" of dental interns' experiences, and through qualitative research, we understand the "how" and the "why." The diverse methodologies used in the dental internship literature provide a fuller picture of dental interns' experiences and the impact of these internships from which we can establish a solid body of knowledge.

## Research implications

The findings of this research have several implications. First, the results confirm the significance of the activities that dental internships involve, such as clinical rotations, research, community work, and workshops and conferences. The participants in this research reported that these activities contributed to their professional, personal, and social development. This suggests that it is important to retain and enhance these activities in internship programs to ensure that internships provide interns with impactful learning experiences. This can be achieved by improving the diversity and quality of clinical rotations, providing ample opportunities for research and research mentorship, growing the community service work done by

interns, and running additional workshops and conferences on topics that support interns' professional growth. The optimization of dental internship activities will help guarantee that these internships satisfy dental interns' educational needs and contribute to their overall development.

Second, the findings reported here reveal that dental interns experience autonomy during their internships, especially concerning research and clinical practice. It is thus important to recognize and strengthen the autonomy that internships allow interns by, for example, cultivating an environment in which interns make decisions independently, are accountable for their work, and contribute to their professional roles. Emphasizing autonomy during internships can lead to greater professional development, higher general satisfaction, and greater confidence. The priority for internships should be empowering interns to make decisions, which includes giving them the skills, knowledge, and support they need to make informed clinical and research choices. An internship environment where the input of interns is valued and their contribution to decision-making is actively encouraged will help interns to develop professionally and will equip them with the full suite of abilities they need to practice dentistry independently.

Finally, given the finding that dental interns benefit from the greater freedom and time they have during internships, dental education programs should acknowledge the value of supporting the well-being of students by nurturing a healthy work-life balance. Educational institutions can support well-being by implementing strategies that help students to manage their workload and the requirements of their degrees. These strategies include establishing realistic expectations and outcomes, providing sufficient support and resources, and encouraging students to devote time to self-care. An inclusive and supportive environment where the needs and interests of students are recognized will increase satisfaction and success among the student body and reduce burnout.

Combining qualitative and quantitative research methods is a valuable approach to exploring dental interns' experiences. Based on the the examination conducted on the existing literature findings in this research, it is recommended that a mixed-methods approach be adopted in future studies on dental internships to deliver a fuller understanding of this topic.

## Research limitations

This study has several limitations. First, the results of the study are derived from the analysis of data collected exclusively from dental interns from two Saudi Arabian universities, potentially limiting the generalizability of the findings to different contexts. Data collected from a wider variety of subjects, such as dental interns in other dental programs, would increase the findings' generalizability. Collecting data from participants from different settings and institutions would also produce a more thorough picture of the experiences of dental interns. Second, this study used just two instruments to collect data: semi-structured interviews and diaries. No direct observation was used to collect data, which would have strengthened the design of this research. Direct observation could have provided a more objective view of the behaviors, experiences, and interactions of interns that could then have complemented the subjective data that the diaries and interviews provided. Third, a further limitation of this research is that it did not focus on the negative aspects of dental interns' experiences, meaning that this study may produce an incomplete picture of dental interns' experiences and dental internships in general. Fourth, this research did not consider or explore the roles of supervisors from the perspective of dental interns. Fifth, the lack of quantitative data reported in this work limited the researchers' ability to determine the correlations between variables and the significance of the themes identified. For example, this research was unable to assess if there was a statistical

difference between the experiences of male and female dental interns. The combination of qualitative and quantitative data would produce a more detailed picture of the subject of this research.

## Future research

Given the broad scope of the topic of dental internships and the experiences of interns, there are many avenues of enquiry into this subject for future studies to explore. Importantly, it is vital to verify the findings of the existing literature through additional research. Moreover, it would be extremely beneficial to our understanding of dental internships if future research examined different aspects of the internship landscape, such as the specific needs, difficulties, expectations, concerns and negative experiences of dental interns. To fully understand the experiences of dental interns and dental internships in general, future research should also explore the roles played by supervisors from the perspective of interns. Additionally, future research should adopt a mixed-methods approach, using both quantitative and qualitative data to paint a more detailed picture of dental internships. A mixture of data types would facilitate a more in-depth analysis of the experiences of interns and produce a more comprehensive view of internships.

## Conclusion

This qualitative research illuminates Saudi Arabian dental interns' experiences. The analysis of the data revealed the core themes of "activities," "autonomy," and "transition to a balanced life." It was found that dental interns actively participate in a range of activities, in particular, clinical rotations, research, community work, and workshops and conferences, all of which contribute to their development personally, socially, and professionally. Moreover, the study finds that interns experience newfound autonomy during their internships, especially in research and clinical work, further enhancing their professional growth. Through the investigation of dental interns' experiences, this research also goes further than the existing literature by revealing that internships represent the transition from a demanding, hectic undergraduate life to a more balanced, freer lifestyle where interns have the time to pursue interests, friendships, and self-care.

The findings of this research suggest that it is vital to retain and strengthen internship activities including clinical rotations, research opportunities, community work, and workshops and conferences to enhance interns' professional, social, and personal development. Moreover, the professional growth and confidence of interns can be supported by providing an environment that nurtures their autonomy and empowers them to make decisions independently. Also, the importance of a healthy work-life balance should be acknowledged by institutions that provide dental programs. This balance can be achieved by implementing strategies to help undergraduates manage their workload and engage in self-care. Finally, the implications of this study may help support the professional, personal, and social development of interns and fully equip them for their work as dental practitioners.

## Author Contributions

**Conceptualization:** Mohammed Mahmoud Sarhan, Maram Ali M. Alwadi, Saleha Ali Alzahrani, Saad Mahrous Hasubah, Reem Hussain Alhammad, Ali Muhammed Alhussain, Leen Ahmed Qarras, Shadan Hani Sharbib.

**Data curation:** Mohammed Mahmoud Sarhan, Maram Ali M. Alwadi.

**Formal analysis:** Mohammed Mahmoud Sarhan, Saleha Ali Alzahrani, Saad Mahrous Hasubah, Reem Hussain Alhammad, Ali Muhammed Alhussain, Leen Ahmed Qarras, Shadan Hani Sharbib.

**Funding acquisition:** Maram Ali M. Alwadi.

**Methodology:** Mohammed Mahmoud Sarhan, Maram Ali M. Alwadi, Saleha Ali Alzahrani, Reem Hussain Alhammad, Ali Muhammed Alhussain, Leen Ahmed Qarras, Shadan Hani Sharbib.

**Writing – original draft:** Mohammed Mahmoud Sarhan, Maram Ali M. Alwadi, Saleha Ali Alzahrani, Saad Mahrous Hasubah, Reem Hussain Alhammad, Ali Muhammed Alhussain, Leen Ahmed Qarras, Shadan Hani Sharbib.

**Writing – review & editing:** Mohammed Mahmoud Sarhan, Maram Ali M. Alwadi, Saleha Ali Alzahrani, Saad Mahrous Hasubah, Reem Hussain Alhammad, Ali Muhammed Alhussain, Leen Ahmed Qarras, Shadan Hani Sharbib.

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
