## [Decision Letter · Decision Letter 0]

26 Apr 2024

PONE-D-24-05685A Qualitative Investigation of Dental Internships in Saudi Arabia: Exploring the Experiences of Dental InternsPLOS ONE

Dear Dr. Alwadi,

Thank you for submitting your manuscript to PLOS ONE. After careful consideration, we feel that it has merit but does not fully meet PLOS ONE’s publication criteria as it currently stands. Therefore, we invite you to submit a revised version of the manuscript that addresses the points raised during the review process.

**ACADEMIC EDITOR: ** Thank you for submitting the manuscript. After the evaluators' considerations, I encourage them to make major corrections.

We look forward to receiving your revised manuscript.

Kind regards,

Manoelito Ferreira Silva Junior, Ph.D.

Academic Editor

PLOS ONE

 [This research project was supported by a grant from the “Research Center of the Female Scientific and Medical Colleges”, Deanship of Scientific Research, King Saud University.].  

5. In the online submission form, you indicated that [The data that support the findings of this study are available from the corresponding author.]. 

Additional Editor Comments:

Dear Maram Alwadi,

According to the reviewers' observations, I ask that they make the changes and send us a new version of the manuscript.

Reviewers' comments:

Reviewer's Responses to Questions

**Comments to the Author**

1. Is the manuscript technically sound, and do the data support the conclusions?

Reviewer #1: Partly

Reviewer #2: Partly

2. Has the statistical analysis been performed appropriately and rigorously? 

Reviewer #1: N/A

Reviewer #2: N/A

3. Have the authors made all data underlying the findings in their manuscript fully available?

Reviewer #1: Yes

Reviewer #2: Yes

4. Is the manuscript presented in an intelligible fashion and written in standard English?

Reviewer #1: Yes

Reviewer #2: Yes

5. Review Comments to the Author

Reviewer #1: Abstract: The presented section is not balanced. They present little information about methodology, and many conclusions not directly related to the objective. Authors should rebalance their abstracts.

Background: The authors could be more objective in this section. A lot of information is presented that does not help the reader in understanding the problem or aim studied. Affirmation presented in lines 135-137 need references; The hypothesis or assumption investigated are not clear.

Methodology: Research undersection could receive part of background. If the population of study is known (dental interns who had earned bachelor’s degrees in dentistry from two universities in Saudi Arabia), why was snowball technique applied?

As presented in 230 and 231, a qualitative study demands a “literature review or existing theory” to interpret data processed with thematic analysis. However, authors were not clear in the method or background sections. I encourage authors to create an undersection with literature indicators or framework theory used to interpret qualitative data.

There is nor a statement locating the researcher culturally or theoretically. Influence of the researcher on the research, and vice- versa, was not addressed. The methodology section could benefit it authors clarify the philosophical perspective that guide the study.

After this, reviewers would answer the following question: “Is there congruity between the stated philosophical perspective and the research methodology?”

Results are well presented, including students’ speeches. However, the authors did not present the characterization of students included in the study.

Reviewer #2: To improve knowledge in this field, it would be important to include negative feelings regarding internship. Considering only the positive aspects offers a superficial view of the topic. It would also be important to analyze the role of supervisors/tutors in students' opinions. Another aspect to be taken into account is whether there is any influence of gender on the responses.

6. PLOS authors have the option to publish the peer review history of their article (what does this mean?). If published, this will include your full peer review and any attached files.

Reviewer #1: **Yes: **Álex Moreira Herval

Reviewer #2: No

---

## [Author Response · Author response to Decision Letter 0]

7 May 2024

Dear Editor,

I hope you are doing well.

We would like to let you know that our responses to the reviewers have been attached within the submission.

We hope this version of the manuscript meets your expectations.

Sincerly, 

Maram Alwadi

---

## [Decision Letter · Decision Letter 1]

28 May 2024

PONE-D-24-05685R1A Qualitative Investigation of Dental Internships in Saudi Arabia: Exploring the Experiences of Dental InternsPLOS ONE

Dear Dr. Alwadi, Thank you for submitting your manuscript to PLOS ONE. After careful consideration, we feel that it has merit but does not fully meet PLOS ONE’s publication criteria as it currently stands. Therefore, we invite you to submit a revised version of the manuscript that addresses the points raised during the review process.

We look forward to receiving your revised manuscript.

Kind regards,

Manoelito Ferreira Silva Junior, Ph.D.

Academic Editor

PLOS ONE

Journal Requirements:

**Additional Editor Comments:**

All the efforts made by the authors to respond to all the items requested by the reviewers, after a new round of evaluation, some points still deserve to be highlighted and therefore, I request further minor corrections below.

Reviewers' comments:

Reviewer's Responses to Questions

**Comments to the Author**

1. If the authors have adequately addressed your comments raised in a previous round of review and you feel that this manuscript is now acceptable for publication, you may indicate that here to bypass the “Comments to the Author” section, enter your conflict of interest statement in the “Confidential to Editor” section, and submit your "Accept" recommendation.

Reviewer #3: All comments have been addressed

2. Is the manuscript technically sound, and do the data support the conclusions?

Reviewer #3: Yes

3. Has the statistical analysis been performed appropriately and rigorously? 

Reviewer #3: N/A

4. Have the authors made all data underlying the findings in their manuscript fully available?

Reviewer #3: Yes

5. Is the manuscript presented in an intelligible fashion and written in standard English?

Reviewer #3: Yes

6. Review Comments to the Author

Reviewer #3: For author and editor,

Review of the submission Manuscript Number PONE-D-24-05685R1: "A Qualitative Investigation of Dental Internships in Saudi Arabia: Exploring the Experiences of Dental Interns".

Thank you very much for submitting the manuscript to the PLOS ONE!

The revisions that you made to the manuscript are very effective in addressing the remaining concerns. Following the review of the previous version, considering the scientific merit, the originality of the manuscript and the reading of the new version, the recommendation is "MINOR REVISIONS".

Congratulation one your fine research!

Cordially,

7. PLOS authors have the option to publish the peer review history of their article (what does this mean?). If published, this will include your full peer review and any attached files.

Reviewer #3: **Yes: **Pablo Guilherme Caldarelli, Ph.D.

---

## [Author Response · Author response to Decision Letter 1]

3 Jun 2024

Dear editor and reviewers, 

We would like to let you know that our responses are attached within the submission. We hope this version of manuscript meets your expectation. 

Best regards,

---

## [Decision Letter · Decision Letter 2]

2 Jul 2024

A Qualitative Investigation of Dental Internships in Saudi Arabia: Exploring the Experiences of Dental Interns

PONE-D-24-05685R2

Dear Dr. Maram Alwadi,

We’re pleased to inform you that your manuscript has been judged scientifically suitable for publication and will be formally accepted for publication once it meets all outstanding technical requirements.

Kind regards,

Manoelito Ferreira Silva Junior, Ph.D.

Academic Editor

PLOS ONE

Additional Editor Comments (optional):

I would like to thank the authors for their meticulous work in answering all the evaluators’ points and questions. In this sense, I consider the current version approved for publication.

Reviewers' comments:

Reviewer's Responses to Questions

**Comments to the Author**

1. If the authors have adequately addressed your comments raised in a previous round of review and you feel that this manuscript is now acceptable for publication, you may indicate that here to bypass the “Comments to the Author” section, enter your conflict of interest statement in the “Confidential to Editor” section, and submit your "Accept" recommendation.

Reviewer #3: All comments have been addressed

2. Is the manuscript technically sound, and do the data support the conclusions?

Reviewer #3: Yes

3. Has the statistical analysis been performed appropriately and rigorously? 

Reviewer #3: N/A

4. Have the authors made all data underlying the findings in their manuscript fully available?

Reviewer #3: Yes

5. Is the manuscript presented in an intelligible fashion and written in standard English?

Reviewer #3: Yes

6. Review Comments to the Author

Reviewer #3: For author and editor,

Review of the submission PONE-D-24-05685R2: "A qualitative investigation of dental internships in Saudi Arabia: Exploring the experiences of dental interns".

Thank you very much for submitting the manuscript to the PLOS ONE!

The revisions that you made to the manuscript are very effective in addressing the remaining concerns. Following the review of the previous version, considering the scientific merit, the originality of the manuscript and the reading of the new version, the recommendation is "ACCEPT ".

Congratulation one your fine research!

Cordially,

7. PLOS authors have the option to publish the peer review history of their article (what does this mean?). If published, this will include your full peer review and any attached files.

Reviewer #3: **Yes: **Pablo Guilherme Caldarelli

---

## [Editor Report · Acceptance letter]

4 Jul 2024

PONE-D-24-05685R2 

PLOS ONE

Dear Dr. Alwadi, 

I'm pleased to inform you that your manuscript has been deemed suitable for publication in PLOS ONE. Congratulations! Your manuscript is now being handed over to our production team.

Kind regards, 

on behalf of

Dr. Manoelito Ferreira Silva Junior 

Academic Editor

PLOS ONE